

# Experimental moose reduction lowers wolf density and stops decline of endangered caribou

Robert Serrouya[1,2], Bruce N. McLellan[1,3], Harry van Oort[1], Garth Mowat[4,5] and Stan Boutin[6]

[1] Columbia Mountains Caribou Research Project, Revelstoke, British Columbia, Canada
[2] Alberta Biodiversity Monitoring Institute, University of Alberta, Edmonton, Alberta, Canada
[3] Research Branch, Ministry of Forests, Lands, and Natural Resource Operations, D'Arcy, British Columbia, Canada
[4] Natural Resource Science Section, Ministry of Forests, Lands, and Natural Resource Operations, Nelson, British Columbia, Canada
[5] Department of Earth and Environmental Sciences, University of British Columbia Okanagan Campus, Kelowna, British Columbia, Canada
[6] Department of Biological Sciences, University of Alberta, Edmonton, Alberta, Canada

Corresponding author
Robert Serrouya,
serrouya@ualberta.ca

## ABSTRACT

The expansion of moose into southern British Columbia caused the decline and extirpation of woodland caribou due to their shared predators, a process commonly referred to as apparent competition. Using an adaptive management experiment, we tested the hypothesis that reducing moose to historic levels would reduce apparent competition and therefor recover caribou populations. Nested within this broad hypothesis were three specific hypotheses: (1) sport hunting could be used to substantially reduce moose numbers to an ecological target; (2) wolves in this ecosystem were primarily limited by moose abundance; and (3) caribou were limited by wolf predation. These hypotheses were evaluated with a before-after control-impact (BACI) design that included response metrics such as population trends and vital rates of caribou, moose, and wolves. Three caribou subpopulations were subject to the moose reduction treatment and two were in a reference area where moose were not reduced. When the moose harvest was increased, the moose population declined substantially in the treatment area (by 70%) but not the reference area, suggesting that the policy had the desired effect and was not caused by a broader climatic process. Wolf numbers subsequently declined in the treatment area, with wolf dispersal rates 2.5× greater, meaning that dispersal was the likely mechanism behind the wolf numerical response, though reduced recruitment and starvation was also documented in the treatment area. Caribou adult survival increased from 0.78 to 0.88 in the treatment area, but declined in the reference. Caribou recruitment was unaffected by the treatment. The largest caribou subpopulation stabilized in the treatment area, but declined in the reference area. The observed population stability is comparable to other studies that used intensive wolf control, but is insufficient to achieve recovery, suggesting that multiple limiting factors and corresponding management tools must be addressed simultaneously to achieve population growth.

## INTRODUCTION

When species colonize new areas, the consequences for native organisms can be profound, often with negative impacts caused by competition or predation. An exotic predator can have dramatic effects on native prey (*Smith & Quin, 1996*), particularly on islands where prey have evolved few anti-predator strategies (*Sinclair et al., 1998*). Similarly, extreme forms of interference competition can have pronounced and obvious impacts, such as the invasion of the Eurasian zebra mussel (*Dreissena polymorpha*) into North America where it now dominates available substrate and smothers native bivalves (*Ricciardi, Neves & Rasmussen, 1998*). In both cases, the ecological interactions can be severe but straightforward to document. Interactions involving indirect processes can be more difficult to confirm because they are not well explained simply by tracking the abundance of individuals. One such process is apparent competition (*Holt, 1977*), which can occur when a novel prey species colonizes a new area, stimulating an increase in the abundance of one or more predator species. The novel prey need not be an introduced exotic, but may be expanding its range either because of natural or anthropogenic factors (*Dawe, Bayne & Boutin, 2014*). A secondary, but native prey may then become victim of apparent competition, usually because it is less fecund or less able to escape predation than the novel prey. The secondary prey can be driven to extinction because there is little or no feedback between secondary prey abundance and predator numbers, given that predators are sustained by the more abundant novel prey (*Holt, 1977*; *Latham et al., 2011*). Identifying this mechanism can be difficult because the cause of the secondary prey's decline could be confused with other indirect interactions such as exploitative competition.

Woodland caribou (*Rangifer tarandus caribou*) represent a classic case of apparent competition, especially the endangered ecotype of mountain caribou that inhabit the interior rain forests of British Columbia and Idaho. Increases in moose (*Alces alces*) and white-tailed deer (*Odocoileus virginianus*) are leading to the unsustainable predation rates on caribou (*Seip, 1992*; *Latham et al., 2011*). Recovery options for mountain caribou can be summarized into three interrelated approaches (*Seip, 2008*). The first is to reduce or eliminate forest harvesting in caribou range because this activity increases forage for moose and deer. Forest harvesting also reduces the abundance of arboreal lichens that are the primary food for mountain caribou during winter. However, reducing forest harvesting will not prevent the imminent extinction of many caribou subpopulations (*Wittmer, Ahrens & McLellan, 2010*) because of the existing legacy of forestry; it will take decades for natural succession to reduce forage for moose and deer. The second option is to directly reduce predator numbers. In numerous locations this approach has been shown to increase caribou vital rates and population trend (*Bergerud & Elliott, 1998*; *Seip, 1992*; *Hayes et al., 2003*; *Hervieux et al., 2014*). Predator reduction, however, must be continuous because if the treatment is stopped, predator numbers recover quickly, as long as their primary prey

are still abundant (*Ballard, Whitman & Gardner, 1987*; *Hayes et al., 2003*). Predator control is also much less acceptable to the public than it was in the past (*Orians et al., 1997*, but see *Boertje, Keech & Paragi, 2010*). The third option involves reducing the primary prey that supports predator populations, under the premise that this action will indirectly reduce predator numbers.

Recent recovery actions across the range of mountain caribou have included protecting >2 million hectares of old-growth forest from logging, closing areas to snowmobiling, and adhering to a minimum distance between mechanized recreation and observed caribou. Yet because these management actions do not deal with proximate limiting factors, a population response from caribou has not been observed, even in herds that live in parks and have seen minimal disturbance on their range (*Hebblewhite, White & Musiani, 2010*; *Serrouya & Wittmer, 2010*). Viability analyses suggest that under current conditions, without any additional habitat degradation, many mountain caribou populations are on a trajectory to extinction (*Wittmer, Ahrens & McLellan, 2010*). Clearly this means that direct management of animal populations is needed, and such actions should be implemented across broad spatial scales (*Carpenter et al., 1995*) that large mammal predator–prey systems are known to encompass (*Hayes et al., 2003*; *Mosnier et al., 2008*).

The hypothesis we tested was whether substantially reducing moose numbers, the wolves' (*Canis lupus*) primary prey in this ecosystem, to an ecological target (*Serrouya et al., 2011*) would reduce wolf populations and thus positively affect caribou population growth. The ecological target was based on the estimated number of moose that would exist if there had been no logging, which implied a reduction of *c.* 70% over contemporary numbers (*Serrouya et al., 2011*). Nested within our broad hypothesis were three more specific hypotheses, each contingent on the previous one: (1) sport hunting could be used to substantially reduce moose numbers; (2) wolves in the Columbia ecosystem were primarily limited by moose abundance; and (3) mountain caribou were partially limited by wolf predation. Theoretical underpinnings of the broad hypothesis were explored in *Serrouya et al. (2015)* who found that the rate and intensity of reducing invading prey would have a major influence on the native prey. In this paper we focus on empirical data including before-after control-impact (BACI) comparisons of large areas that were subject to the moose reduction, with a spatial reference area where moose were not reduced, and on the vital rates of the large mammals under study (caribou, moose, and wolves).

The ability to use sport hunting to reduce ungulate populations is important because hunter access and the fecundity rate of the prey species make some cervid populations resilient to increased hunting (*Brown et al., 2000*; *Lebel et al., 2012*; *Simard et al., 2013*). To properly test this hypothesis, we had to establish whether any change in moose abundance was caused by the change in harvest policy, or a broader ecological process such as climate/weather or ecosystem change that can also influence ungulate populations (*Post & Stenseth, 1998*). The spatial reference area where moose hunting was not increased helped to resolve these potential confounds.

The hypothesis that wolves were primarily limited by moose abundance simply predicts that reducing moose will reduce wolf abundance. Descriptive studies from within the study area suggested that wolf diets were dominated by moose (*Stotyn, 2008*), and across

a variety of ecosystems there is a broad relationship linking ungulate biomass to wolf abundance (*Fuller, Mech & Cochrane, 2003*). However, wolf populations lag, sometimes by many years, in response to a decline in their primary prey (*Mech, 1977*; *Gasaway et al., 1983*). In such a case, the ratio of wolves to prey would increase, at least temporarily, which could be detrimental to caribou because they would become a higher proportion of available prey. There is increasing evidence that higher predator to prey ratios translate to higher per capita predation rates (*Vucetich et al., 2011*). In addition, wolf populations may not track the availability of moose biomass, but instead respond to the abundance of vulnerable (old) moose (*Peterson et al., 1998*). These factors may result in an equivocal relationship between wolf abundance and the moose reduction treatment, particularly at a local scale.

The final hypothesis was that mountain caribou were limited in part by wolf predation (*Wittmer et al., 2005*), and therefore our prediction was that reduced wolf numbers would at least increase caribou survival and reduce their rate of decline compared to before the treatment and to the spatial reference area. This hypothesis has received support across many woodland caribou subpopulations, either through manipulation (*Seip, 1992*; *Hayes et al., 2003*; *Hervieux et al., 2014*) or mensurative experiments (*Seip, 1992*; *Rettie & Messier, 1998*; *McLoughlin et al., 2003*; *Wittmer, Sinclair & McLellan, 2005*; *Latham et al., 2011*). We also predicted that caribou recruitment would increase following the treatment, recognizing that caribou calves might be killed by bears or meso-predators to a greater extent than wolves (*Adams, Singer & Dale, 1995*; *Gustine et al., 2006*). Again, these predictions depended on the previous hypothesis, where a reduction in wolves could be achieved by reducing moose.

## STUDY AREA

The study was located within two major mountain ranges in the interior of British Columbia, the Columbia and Cariboo Mountains. The treatment area was 6,500 km$^2$ whereas the reference area was 11,500 km$^2$, and they were separated by the Monashee Mountains, a sub-range of the Columbia's with a maximum elevation of 3,274 m. All these areas were windward ranges of the Rocky Mountains that had wet climates dominated by interior rain-forests. Both areas were rugged and remote. Half of the treatment area was on the west side of Lake Revelstoke with only boat access (Fig. 1), and thus little human presence. In the reference area there was a major highway and railway, whereas in the treatment area there was one dead-end highway with relatively little traffic. Warm summers and cool, wet winters with excessive snowfall (>20 m) are typical in the central portion of these ranges at mid elevations (1,800 m a.s.l.) where caribou spend most of the winter. In valley bottoms (400–500 m) snowfall averages 396 cm ($n = 100$ yr, SD $= 120$), which is where most other ungulates and their predators spend the winter. As the snow melts in summer, moose and deer, along with wolves, bears (*Ursus* spp.), and cougars (*Puma concolor*), spread out in the mountains. As a rough ratio of moose to deer abundance, sampling from 17 pellet transects cleared and measured each spring from 2003 to 2011 (*Serrouya et al., 2011*) recorded 969 moose and 61 deer pellet groups (a ratio of 15.9:1).

Below approximately 1,300 m, western redcedar (*Thuja plicata*) and western hemlock (*Tsuga heterophylla*) are the typical climax tree species, whereas above this elevation

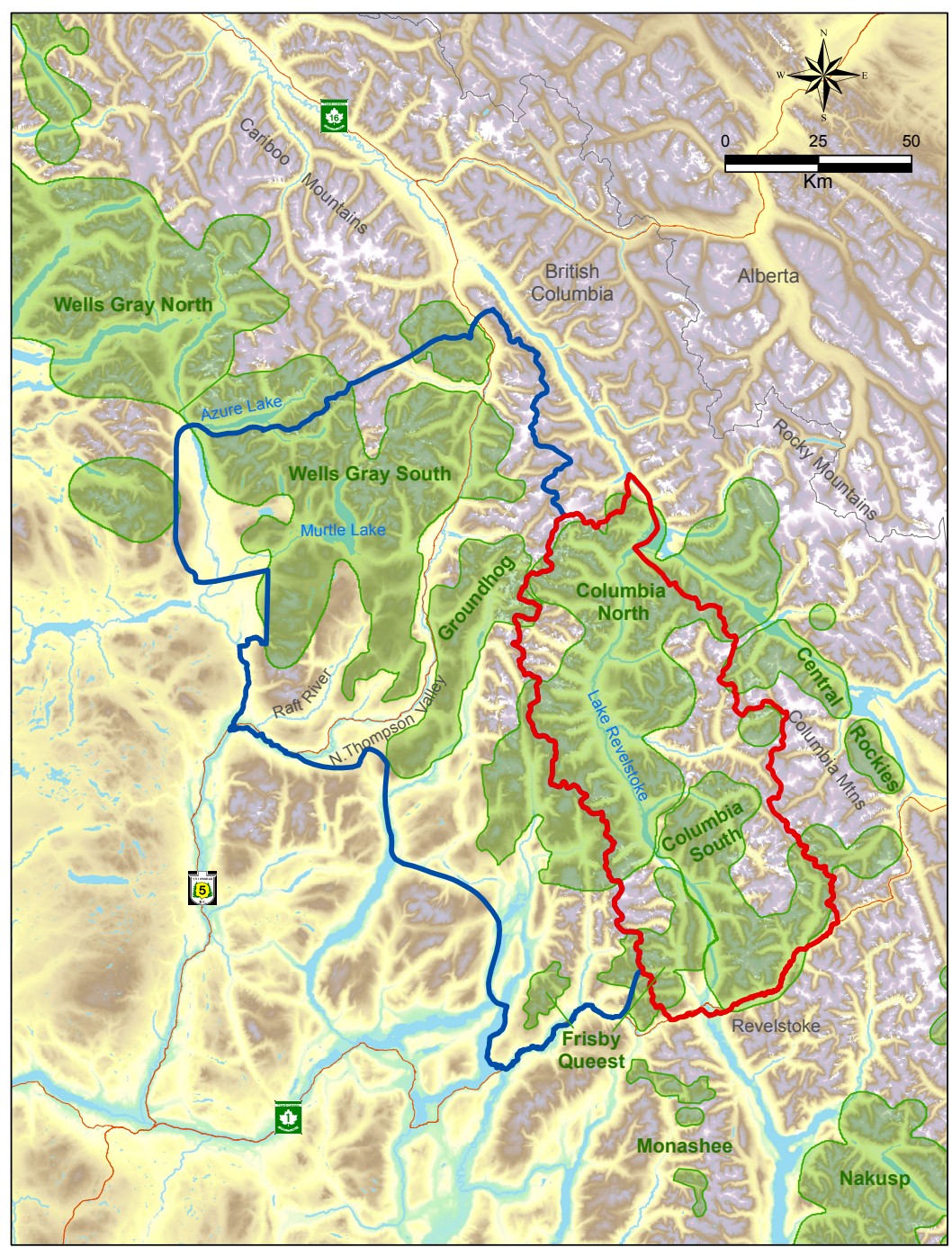

**Figure 1  Map of treatment (red) vs. reference areas (blue).** Caribou subpopulations are shaded green.

the forest transitions to Engelmann spruce (*Picea engelmannii*) and subalpine fir (*Abies lasiocarpa*). Forest age classes are typically bi-modal, with either old (>250 yr) or young (<40 yr) regenerating stands following harvesting, and relatively little (<15%) mid-seral vegetation. Natural shrub openings occur in avalanche paths, marshes, and at higher elevation as the forest transitions to alpine areas.

The caribou subpopulations in the treatment area included Columbia North, Columbia South, and Frisby-Queest. The latter two each numbered less than 50 caribou when the moose reduction treatment began, while Columbia North numbered *c.*150 when the treatment began (*McLellan, Serrouya & Flaa, 2006*). A portion of the Frisby-Queest range was located in the reference area, but the vast majority (>80%) of caribou occurrences from that subpopulation were located in the treatment area (*Apps & McLellan, 2006*). In the reference area, caribou subpopulations were Wells Gray South and Groundhog, which numbered *c.* 120 and 20, respectively, at the beginning of the experiment. Although *Wittmer et al. (2005)* considered Wells Gray North and South to be one subpopulation, more recent analyses revealed limited demographic exchange between these areas (*Van Oort, McLellan & Serrouya, 2011*; *Serrouya et al., 2012*), so the comparison was limited to the Wells Gray South portion (Fig. 1) of the larger Wells Gray subpopulation.

## METHODS

To estimate population size, trend, recruitment, survival, mortality causes and dispersal rates, animals were captured and fitted with VHF or GPS radio collars. Caribou and moose were captured by aerial net-gunning, whereas wolves were captured using leg-hold traps and net-gunning. To reduce confounding effects of age for estimates of vital rates, we avoided capturing animals that were <2 years old (and to avoid risks with radio collaring). Net-gunning was conducted in winter when snow facilitated tracking and minimized the risk of injury to animals, whereas leg-hold traps were used in summer only. Captures adhered to BC Provincial Government and University of Alberta animal care protocols (permit # VI08-49757, and 690905, 2004-09D, 2005-19D).

Animals were monitored every two to four weeks from fixed-wing aircraft using VHF telemetry. If an animal was not found during a monitoring session, the pilot (D Mair, Silvertip Aviation) scanned for these animals while *en route* to other projects centered 150 to 300 km away in BC and Alberta, flying at a high altitude (>2,500 m) to maximize collar detection. We occasionally searched a 50–100 km buffer around the study area using a meandering flight path to try to locate missing animals.

### Study design and response metrics

The design was based on an ecosystem-level perturbation intended to reduce moose populations in one area (treatment) and compare the results to the reference area where no attempt was made to reduce the moose population. In the treatment area, a 10-fold increase in the number of moose hunting permits began in 2003 (Fig. 2; *Serrouya, McLellan & Boutin, 2015*), but no major change in policy occurred in the reference area. In the treatment and reference areas we estimated moose survival and population trend, wolf survival and dispersal, and caribou survival, abundance, trend, and recruitment. In the

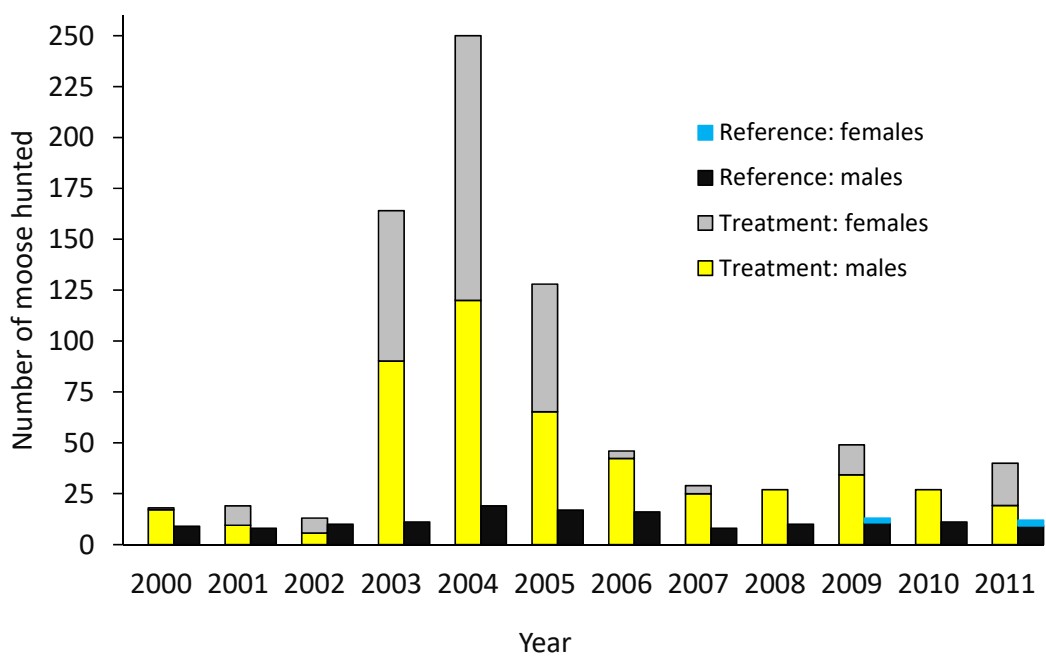

**Figure 2  Number of moose harvested in the treatment and reference areas.** The moose reduction treatment began in 2003. From *Serrouya, McLellan & Boutin (2015)*.

treatment area alone, we also estimated wolf abundance, trend, and recruitment (Table 1). Caribou monitoring began in 1992 (summarized in *Wittmer et al., 2005*), so we compared population parameters before and after the moose reduction treatment began, and against the spatial reference area, conforming to a BACI design. Moose abundance and survival estimates began in 2003, but population trend based on hunter harvest data could be estimated in both areas prior to this date. Wolf survival and dispersal comparisons began in 2004, but abundance estimates began in 2007. For moose and wolves, we present animal abundance and density, because density fluctuates within a year as the available habitat changes 3-fold in summer vs. winter when snow restricts the distribution of moose and wolves to the valley bottom. Winter densities can be obtained by dividing abundance by *c.* 1,100 km$^2$, but summer densities are roughly 3-fold less as snow melts and animals disperse into adjacent mountains (*Serrouya et al., 2011*).

## Moose abundance, trend, and survival

Methods to estimate moose abundance and trend were described in *Serrouya et al. (2011)*. Briefly, in the treatment area moose abundance was estimated using stratified random block aerial surveys (*Gasaway et al., 1986*) and trend was monitored using annual pellet transects (*Serrouya et al., 2011*). Catch per unit effort (CPUE) hunting data were calibrated against these values (*Serrouya, McLellan & Boutin, 2015*) and compared to CPUE data from the reference area. In the reference area, moose abundance was estimated in 2007 in the northern third of the area (see reference area in *Serrouya, McLellan & Boutin, 2015*), and in the Raft River in 2009 (*Klafki, Poole & Serrouya, 2009*). These two point estimates were not used to estimate population trend, which was estimated using CPUE data.

**Table 1  Response metrics within the treatment and reference areas (Yes (Y) or No (N)).** Y in brackets also indicates whether data exist before the moose reduction treatment began (i.e., BACI).

| Metric | Treatment | Reference |
| --- | --- | --- |
| Wolf survival and dispersal | Y | Y |
| Wolf abundance | Y | N |
| Wolf trend | Y | N |
| Wolf recruitment | Y | N |
| Moose trend | Y (Y) | Y (Y) |
| Moose abundance | Y (Y) | Y |
| Moose survival | Y | Y |
| Caribou abundance and trend | Y (Y) | Y (Y) |
| Caribou survival | Y (Y) | Y (Y) |
| Caribou recruitment | Y (Y) | Y (Y) |

We calculated daily survival as 1—(no. deaths)/(days monitored) for 2 risk periods: winter (i.e., Nov–Apr) and summer (May–Oct). To produce seasonal survival rates for winter and summer, we exponentiated daily survival rates by 181.25 for winter and 184 for summer (the no. of days in each period). We then calculated annual survival as the product of the 2 seasonal risk periods (*Heisey & Fuller, 1985*). Several authors argue for alternative approaches such as cumulative incidence functions (*Heisey & Patterson, 2006*; *Murray, 2006*), but because we were not using any covariates to explain variation in survival, the Heisey–Fuller method was appropriate and is still commonly used in survivorship studies (*Sparkman, Waits & Murray, 2011*). To obtain 95% confidence intervals, we bootstrapped the distribution of animals 3,000 times and used the percentile method. *P*-values for comparisons between areas were based on matching each bootstrap iteration from the treatment and reference, counting the number of times the treatment values were greater, and converting this to a percentile.

## Wolf abundance, trend, and vital rates

Wolf survival, dispersal, and cause-specific mortality rates were compared between the treatment and reference (Table 1). These rates were estimated again using the Heisey–Fuller method (*Heisey & Fuller, 1985*), but relative to ungulates, wolf mortality patterns vary unpredictably throughout the year. Therefore, to help reduce potential biases stemming from changing mortality risk throughout the year, we chose 12 risk periods corresponding to each calendar month, which converges to the Heisey–Patterson approach (*Heisey & Patterson, 2006*). We again bootstrapped individual animals to obtain confidence intervals and *p*-values for survival, cause-specific mortality rates, and dispersal rates. We also calculated an effective survival rate by considering a dispersed animal to be "dead" from the study system. Cause-specific mortality was separated into five categories: starvation, road kill, hunting and trapping, predation, and unknown cause.

We defined dispersal as animals leaving the treatment or reference area by at least 50 km and not returning by the time the study ended—which was the same as used in a similar study conducted on the Parsnip caribou herd in central British Columbia (*Steenweg, 2011*). However, because the reference unit was substantially larger than the treatment unit, the

opportunity for dispersal in the reference area could be negatively biased. Therefore, we also quantified a more conservative dispersal rate from the treatment unit, by simulating an 11.3-km buffer around the treatment unit, which made it as large as the reference unit. The end result was that in the treatment area wolves would have to disperse at least 61.3 km from the edge. This adjustment was likely overly conservative, because the treatment area was bounded by large mountains so if a wolf left this area it probably reflected an important decision to expend energy and search for resources in a different area where its primary prey were not declining rapidly. Dispersals included wolves that may have been lost from monitoring but were subsequently found outside the study area (*sensu Steenweg, 2011*; *Webb, Allen & Merrill, 2011*) either because they were harvested or were recaptured by another project. Potential dispersers were also estimated and included animals whose radio-signal was lost before the expected end of the collar's life span (*Mills, Patterson & Murray, 2008*; *Steenweg, 2011*). Annual dispersal rates were calculated independently from the other cause-specific rates because including dispersals would have caused a negative bias in the other cause-specific rates (see equations in *Heisey & Fuller, 1985*), particularly since emigrants are normally right censored from analyses. One annual risk period was used to estimate the dispersal rates.

Wolf abundance and trend were estimated in two ways. From 2008 to 2014 each valley below 1,200 m elevation in the treatment area was surveyed within a short time frame (2–4 days). Surveys began one or two days after sufficient snowfall to allow fresh tracks to be detected and easily counted. Helicopters were used extensively, but ground work was done simultaneously on the east side of Lake Revelstoke. Flight paths focussed on areas where tracks could be easily spotted such as forestry roads, cutblocks and wetlands. Ground transects were surveyed using snowmobiles or trucks along plowed and unplowed roads. An attempt was made to locate all wolves in the survey area by trailing each pack until the group size could be counted or at least estimated from tracks. The 2007 estimate was based on an integrated count from multiple data sources over the second half of the winter, including seven track transects sampled 2–3 times, an aerial survey, ground observations, and GPS telemetry for three of the six packs to define pack boundaries. The aerial survey was not a complete census but focussed on counting members of the packs with collared animals and the three known packs without a collared member (*sensu Hayes et al., 2003*). This estimate was considered a minimum because only known packs with territorial animals were enumerated. When trailing wolves, a minimum estimate was always produced. These estimates were obtained from tracking evidence such as splitting of routes followed, or from visual observations of the pack. A maximum estimate was also recorded which provided an upper limit for each pack. The maximum count was more subjective than the minimum count. Each time we encountered a pack we checked for the presence of a collared wolf to estimate the proportion of wolves missed during our survey. This method did not produce a sufficient sample to calculate a correction factor, but did provide an approximate detection rate for the survey method.

Wolf recruitment was estimated only in the treatment area in 2010 and 2011 and was contrasted between a high and a low moose density zone within the treatment area. Moose density was 2.2-fold higher in the high zone (0.43 vs. 0.20/km$^2$, summer density). The

objective of this comparison was to determine if this difference in moose abundance was enough to observe a response in wolf recruitment (*sensu Messier, 1985*). To estimate recruitment, we placed motion triggered cameras (Reconyx, Inc., Holmen, WI, USA) within home ranges of wolves that were monitored using GPS collars. We focused on three wolf packs with existing or recent telemetry data, which allowed us to estimate the location of denning areas. We also placed cameras along known travel routes that were near den sites. Three to four cameras were placed within each territory, and were occasionally moved to help maximize detection of wolves. Commercial wolf urine and gland lure (Kootenay Brand Lures, Kimberly, BC, Canada) was placed near the cameras to slow wolf movements. Cameras were checked approximately once per month and lures were refreshed during these checks. Our metric of recruitment was the number of different pups recorded in the photographs, as a minimum estimate of the number of pups in the pack.

## Caribou population trends, abundance, and recruitment

Caribou abundance, adult survival and recruitment estimates were compared across treatment and reference areas, and before and after the moose reduction was initiated. Caribou censuses were conducted every two years on average, from March to early April when they were high in the mountains and their tracks in open snowfields made them highly visible. Caribou sightability was positively correlated with snowpack depth ($r_s = 0.96$, $p = 0.002$; *Flaa & McLellan, 2000*). When snow depth exceeds 300 cm at 1,800 m a.s.l. (which occurs most winters), sightability is >90% (*Serrouya et al., 2017*). In the 1990s, a large sample of individuals marked with radio collars allowed researchers to correct for missed animals and estimate precision using program NOREMARK (details in *Wittmer et al., 2005*). However, in years when the radiomarked sample was low (i.e., after 2003), the number of caribou observed was used as the estimate. In these years, caribou were not counted unless the snow depth reached 300 cm. Caribou trend was quantified using the finite rate of change (*Caughley, 1977*), also termed lambda ($\lambda$).

Calves were classified as a percent of the population because it was difficult to obtain adult sex ratios without undue harassment. Ungulate recruitment has high inter-annual variability (*Gaillard, Festa-Bianchet & Yoccoz, 1998*) so there is little reason to believe that serial autocorrelation is important, and we grouped recruitment data before and after the treatment, again reducing the need to correct for autocorrelation. We used a linear mixed-effects model (LME) to test whether recruitment changed after the treatment, by creating a variable with two levels (before, after), and evaluated this effect separately in the treatment and reference area. Caribou subpopulation was specified as the random intercept. Recruitment was converted from a proportion using the logit link for LME analyses. Because some populations declined dramatically over the monitoring period, it may be appropriate to correct for population size when estimating recruitment, so that estimates from populations with very few numbers carry less weight. Therefore, we repeated the previous analysis but weighted the model by population size.

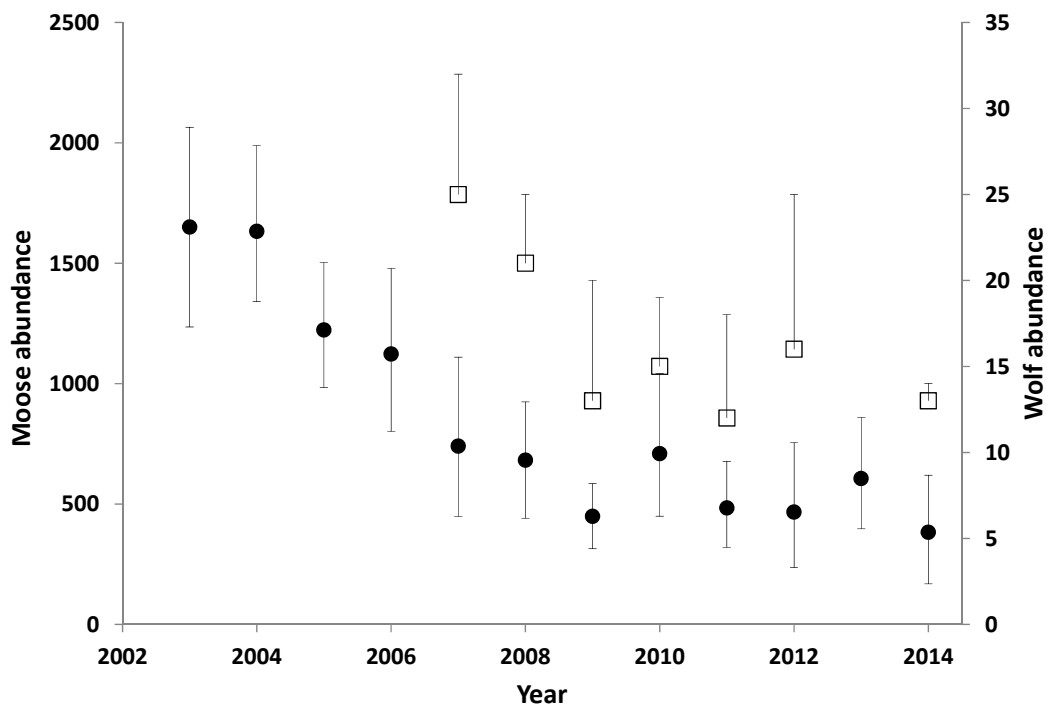

**Figure 3   Moose (circles) and wolf (squares) abundance in the treatment area.** Error bars for the moose estimates represent 90% CIs. The upper error bar for the wolf estimates show the maximum estimates, including a buffer around the treatment area. The square represents a minimum estimate. Between 1994 and 2003 the moose population was estimated to have doubled (*Serrouya et al., 2011*). Data updated from *Serrouya et al. (2015)*.

## RESULTS

### Moose abundance, trend, and survival

The moose population in the treatment area declined by 71% from 2003 when increased hunting began, to 2014 ($\lambda_{annual} = 0.86$; Fig. 3). The average winter density across the treatment area declined from 1.58/km² to 0.44/km² (1,650–466 moose). However, the detectable decline likely began 1–2 years after the treatment was initiated (Fig. 3), so before-after analyses were centered on 2004. The reference area also demonstrated a declining moose trend, but the magnitude was much less than the treatment area. The CPUE data revealed that the slope of decline was more than five times greater in the treatment compared to the reference area (slopes were −6.88, 95% CI[−9.02 to −4.68] compared to −1.32 [−2.46 to −0.265]; Fig. A1). From 2003 to 2009, annual adult moose survival in the treatment area was 0.803 (0.688–0.895, $N = 54$) and 0.878 (0.727–0.972, $N = 13$) in the reference area ($P = 0.18$). Detailed mechanisms of the moose population decline in the treatment area were presented in *Serrouya, McLellan & Boutin (2015)*.

### Wolf dynamics

From 2004–2010, 63 different wolves were captured on 82 separate occasions. Five wolves were not located after capture and collar failure was suspected (they were GPS collars) so were excluded from analyses. Therefore, 58 wolves were available for survival and dispersal

**Table 2** **Annual survival, mortality, and dispersal rates (95% CIs) for wolves in the treatment and reference areas.** Effective survival considers animals that dispersed to equal death from the area. *P*-values were calculated based on a bootstrap comparison of the difference between the two areas.

| Parameter | Treatment | N | Reference | N | P-value |
|---|---|---|---|---|---|
| Survival | 0.726 (0.58–0.85) | 34 | 0.757 (0.56–0.92) | 24 | 0.62 |
| Road kill | 0.024 (0–0.07) | 1 | 0.082 (0–0.21) | 2 | 0.78 |
| Hunt/Trap | 0.105 (0.02–0.21) | 4 | 0.162 (0.04–0.33) | 4 | 0.74 |
| Starvation | 0.063 (0–0.17) | 2 | 0 | 0 | NA |
| Predation | 0.028 (0–0.09) | 1 | 0 | 0 | NA |
| Unknown | 0.053 (0–0.13) | 2 | 0 | 0 | NA |
| Dispersal | 0.221 (0.09–0.39) | 8 | 0.087 (0–0.22) | 2 | 0.08 |
| Dispersal (max)[a] | 0.333 (0.19–0.50) | 13 | 0.239 (0.08–0.43) | 6 | 0.20 |
| Survival (effective) | 0.513 (0.38–0.63) | 34 | 0.586 (0.37–0.77) | 24 | 0.73 |

**Notes.**
[a]Maximum (max) dispersal considers any animal that disappeared while being monitored to be a potential dispersal.

analyses; 34 in the treatment area and 24 in the reference area, with the sex ratio evenly split in both areas. This sampling covered 32.4 monitoring years in the treatment, and 22.2 years in the reference area. There were 12 mortalities of wolves collared in the treatment area. One died after dispersing and was not monitored during the intervening period so it was right censored. Another was a management removal and was also right censored. Eight of 34 (23.5%) wolves dispersed from the treatment area, compared to 2 of 24 (8.3%) in the reference area. If the larger buffer (61.3 km) is considered, then the number of confirmed dispersers from the treatment is reduced from 8 to 7. One additional wolf left the treatment area by crossing west over the Monashee Mountains, but then shed its collar. Because it was only 10 km from the edge of the treatment area it was not counted as a disperser. However, this wolf was likely eating little based on kill rate estimates (*Serrouya, 2013*), and may have crossed the mountains in search of higher moose abundance and could be considered a disperser. If potential dispersers are included (those whose radio signals were lost unexpectedly), then 13 of 34 (38.2%) and six of 24 (25%) wolves dispersed from the treatment and reference areas, respectively. These dispersal values are presented as annual rates in Table 2.

In the reference area there were eight mortalities, including two wolves that dispersed prior to dying (and were right censored for mortality estimation), and no other dispersals were recorded. Thus, in total, eight animals either dispersed or died from the reference area. Survival rates for the two areas were similar (0.726 [0.58–0.85] vs. 0.757 [0.56–0.92]; Table 2). In the treatment area, two wolves from separate packs starved and one was killed by other wolves. Two non-collared wolves from different packs were also found to have starved in the treatment area. One was a pup found at the den site of the collared female that had starved. The other was a subordinate male that starved weeks after its collared pack-mate starved. Human-caused mortality rates (road kills, trapping and hunting) summed to 0.244 in the reference area but 0.129 in the treatment area (Table 2).

There was a minimum of 25 wolves in the treatment area in 2007, but evidence from the integrated count suggests that 32 individuals was more likely and was considered the

**Table 3 Highest number recorded of pups in the treatment area during the summer months using remote cameras in three different wolf territories within two zones of moose density.** Moose density was approximately 0.43/km$^2$ in the High zone (H) and 0.20/km$^2$ in the Low zone (L) (summer density estimates).

| Pack | Zone | Pups | | No. cameras sites | Trap nights | Photos with wolves |
|------|------|------|------|------|------|------|
| | | 2010 | 2011 | | | |
| Gothics | H | 3 | 8 | 7 | 727 | 208 |
| Bigmouth | L | 1[a] | 2 | 6 | 583 | 49 |
| Red Rock | L | 0 | 1[b] | 7 | 706 | 158 |

**Notes.**
[a] This pup was found dead at the den site with its collared mother, which also appeared to starve.
[b] This pup was not recorded in the camera traps but was observed alone while deploying a camera trap along a logging road.

maximum estimate for that year (Fig. 3). Populations were lower and stable from 2009 to 2014, with 2014 having the lowest maximum estimate (14). From 2008 to 2012, all collared packs ($n = 7$) were detected during the survey (no collars were on air in 2014).

Wolf recruitment was higher in the high moose density zone in both 2010 and 2011 (Table 3). Sample sizes were small however (one pack in the high and two packs in the low zone), so no statistics were performed. In 2011, photos from the end of September suggested that the eight pups seen in July for the pack in the high density zone were reduced to four pups. Similarly, the pups seen from the Bigmouth and Red Rock packs earlier in the summer were absent from photos in September that included adult wolves.

## Caribou abundance, recruitment and survival

The two smaller caribou populations in the treatment area (Columbia South and Frisby-Quest) continued to decline after the treatment was initiated (Fig. 4). The rate of decline increased for Columbia South, and remained relatively steady for Frisby-Quest ($\lambda = 0.92 - 0.96$). The growth rate of the largest subpopulation in the treatment area, Columbia North, increased following the moose reduction (Fig. 4), though that increase was primarily because of the 2013 estimate, with 32 additional animals compared to the 2011 estimate. In the reference area, the smaller subpopulation (Groundhog) continued to decline, and the Wells Gray South subpopulation initially appeared to stabilize but a marked reduction was observed in 2013 (Fig. 4). After moose were reduced, adult caribou survival increased from 0.784 to a high of 0.879 in the treatment area (Columbia North subpopulation, pooled $P$-value $= 0.11$), but declined in the reference area (Wells Gray subpopulation; $P < 0.02$; Table 4).

There was no indication that caribou recruitment improved as a result of the treatment (LME $\beta_{before} = 0.456$, SE $= 0.32$, $p = 0.17$, $n = 30$, 3 groups; Fig. 5). The predicted value of recruitment was 15.5% (before) vs. 10.1% (after), and if the analysis was weighted by population size then the values changed little, to 14.3 and 11.5%, respectively. Recruitment improved in the reference area after the treatment began (LME $\beta_{before} = -0.380$, SE $= 0.15$, $p = 0.02$, $n = 20$, 2 groups; 13.1% (before) vs. 18.1% (after); Fig. 5), but when population size was accounted for the difference was negligible (14.8%–16.3%), and non-significant ($p = 0.38$) likely because the high recruitment value (33%; Fig. 5) from Groundhog in 2011 was discounted due to its much reduced population size ($n = 9$; Fig. 4).
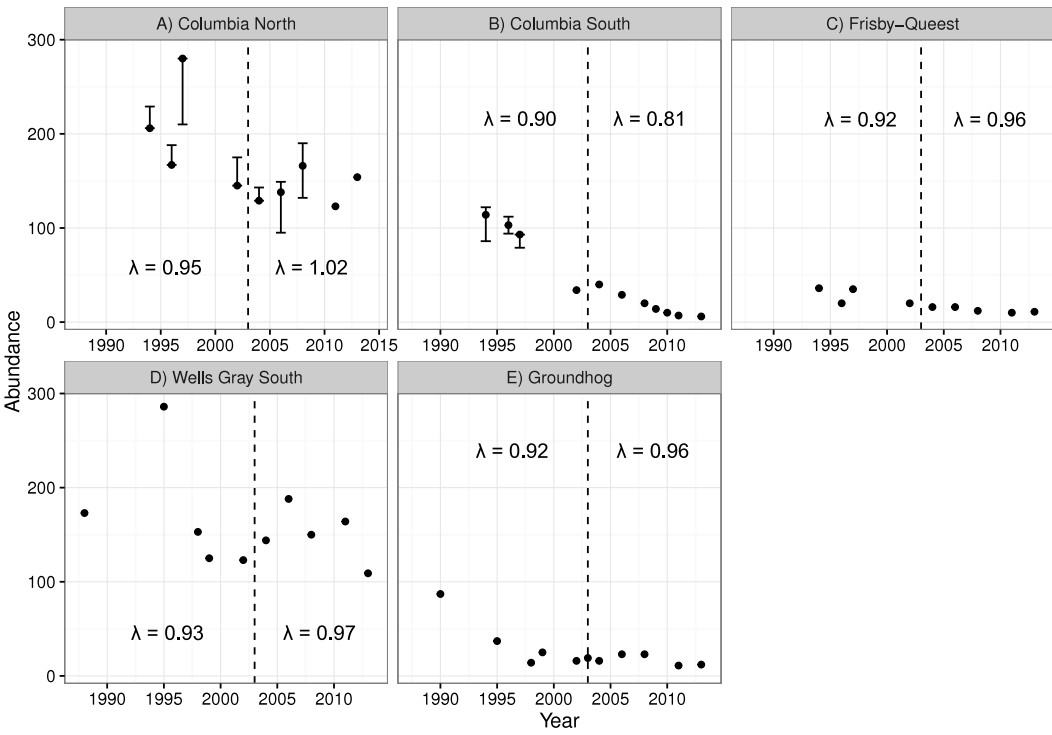

**Figure 4** **Caribou population estimates for five subpopulations, three in the treatment area (A–C; top row) and two in the reference area (D, E; bottom row).** Error bars are 95% CIs. The dashed vertical line represents the beginning of the moose reduction treatment in 2003. Lambda values to the left of the dashed line represent the years 1994–2004 whereas those to the right of the line represent the years 2004–2013. Data updated from *Wittmer et al. (2005)*.

**Table 4** **Survival of radio-marked adult caribou in the treatment and reference area, both before and after the treatment began (pre and post 2004, respectively).** *N* is the number of caribou monitored during the time period. For this comparison, the treatment area included the Columbia North subpopulation, and the reference area was the Wells Gray subpopulation.

| Treatment area | | | Reference area | | |
|---|---|---|---|---|---|
| Time period | Survival | *N* | Time period | Survival | *N* |
| Before treatment | | | | | |
| 1996–2002 | 0.793 (0.697–0.873) | 40 | 1997–2004 | 0.868 (0.801–0.923) | 39 |
| 2002–2004 | 0.784 (0.556–0.944) | 17 | | | |
| After treatment | | | | | |
| 2004–2006 | 0.879 (0.731–1.0) | 14 | 2004–2010 | 0.725 (0.531–0.851) | 15 |
| 2006–2008 | 0.857 (0.676–1.0) | 10 | | | |

# DISCUSSION

The primary hypothesis tested was that reducing moose abundance to an ecological target would reduce the caribou's rate of decline. Within this hypothesis were three nested hypotheses and predictions. The first of these was confirmed: moose were reduced using sport hunting and two lines of evidence support this conclusion. First, the population declined at a rate that was 5-fold greater than in a spatial reference area where the hunting

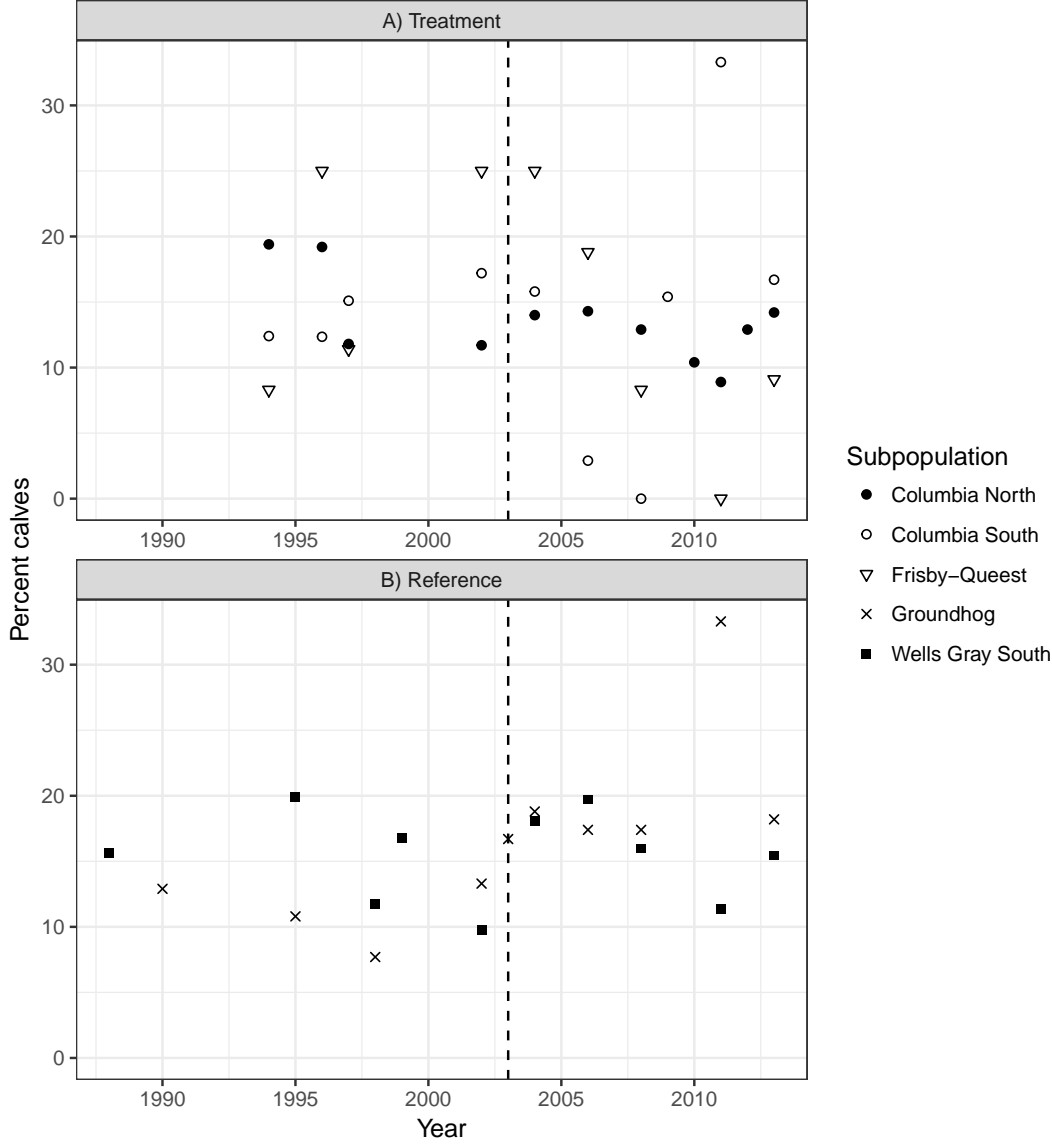

**Figure 5  Caribou recruitment (% calves) within the treatment (A) and reference areas (B).** The vertical line represents when the moose reduction began in the treatment area. Symbols represent separate subpopulations.

kill was not increased. Second, we contrasted the effects of hunting compared to predation and found that both factors contributed to the decline, with the increased hunt initiating the depensatory predation rate observed in the treatment area (*Serrouya, McLellan & Boutin, 2015*). It may seem intuitive that increasing hunting pressure on a large herbivore would reduce its abundance, particularly with more females harvested. However, other North American cervids have been difficult to control using sport hunting (*Brown et al., 2000*; *McDonald, Clark & Woytek, 2007*; *Simard et al., 2013*) because of poor access, urban refuges (*Polfus & Krausman, 2012*), or high fecundity and immigration rates. In one of the few other experimental attempts to reduce an overabundant ungulate, *Simard et al.*
*(2013)* found that white-tailed deer were not reduced in replicated 20-km$^2$ treatments on a predator-free island. Moose are less fecund than white-tailed deer, which may explain the discrepancy between the two systems. Furthermore, our treatment area was 300 times larger, with less chance of immigration because of the closure imposed by rugged mountain ranges.

The second of the nested hypotheses was that wolves were primarily limited by moose abundance, with the prediction that reducing moose would reduce wolf numbers. Ostensibly, this hypothesis appears trivial because of established relationships between ungulate biomass and wolf abundance (*Fuller, Mech & Cochrane, 2003*), but complimentary explanations have been proposed related to the age or vulnerability of moose, or social constraints within wolf packs that limit wolf abundance regardless of food availability (*Messier, 1994*; *Peterson et al., 1998*; *Hebblewhite, 2013*; *Cubaynes et al., 2014*). This hypothesis was also supported, because wolf numbers declined following the moose reduction. The lack of wolf survey data prior to 2007 weakens this conclusion, as does the absence of trend data of wolves in the reference area. Nonetheless, we identified several mechanisms to explain the reduction in wolf abundance. First, the dispersal rate was greater in the treatment area than the reference area, resulting in a relatively low effective survival rate. At 0.51, the effective survival rate was lower than the minimum level (0.64) required to maintain a stable wolf population (*Fuller, Mech & Cochrane, 2003*). These results mirror those of *Steenweg (2011)* who found dispersal to be the primary wolf vital rate affected by a moose reduction treatment in the Parsnip area. The dispersal rate in our treatment area was also greater than what was estimated by *Webb, Allen & Merrill (2011)*, who reported an emigration rate of 0.13 in a population in Alberta. Second, we found some evidence of reduced wolf recruitment in areas with lower moose densities (0.2/km$^2$). *Messier (1985)* also found that at moose densities <0.2/km$^2$, wolves had difficulty recruiting pups. Third, wolf starvation was recorded in the treatment area but not the reference area. Finally, analyses of wolf scat contents showed that wolves were primarily supported by moose in this study area (*Stotyn, 2008*; *Serrouya, 2013*) so a reduction in wolf numbers is consistent with a reduction in their major food source.

Although trappers removed wolves in the area, the overall trapping and hunting rate (0.11) is low compared to other populations. In Alberta, *Webb, Allen & Merrill (2011)* estimated that a harvest rate of 0.34 had no effect on the population trend of wolves. Few wolves were trapped or hunted in the study area because the amount of snowfall confounds trap sets and thick cover obscures visibility for hunting. Therefore, changes in wolf abundance can be attributed to a reduction in food and not intensive human harvest.

The final hypothesis, that reduced wolf numbers would lessen the rate of caribou decline relative to spatial and temporal contrasts, was partially supported. The three smallest caribou populations continued to decline, regardless of whether they were in the treatment or reference area. A number of mechanisms can negatively affect small populations including environmental stochasticity and Allee effects (*Allee, 1931*) resulting from predation, and both of these processes have affected woodland caribou (*Hebblewhite, White & Musiani, 2010*; *McLellan et al., 2010*). We acknowledge that only the largest of three subpopulations subjected to the moose reduction demonstrate improved demographic

trend. However, a parallel study in the Parsnip region showed similar improvement to caribou populations when moose were experimentally reduced (*Steenweg, 2011*; D Heard, 2016, unpublished data).

For the two largest subpopulations (Columbia North and Wells Gray), a modest increase was observed in the treatment area (Columbia North) but a sharp decline in the reference area (Wells Gray). It is tempting to conclude that the experiment was not a success because caribou population growth was not immediate and pronounced in the treatment area. However, our results are similar to other management actions intended to recover declining caribou herds. In Alberta, 841 wolves were removed in a caribou range from 2005 to 2012 and resulted in caribou λ increasing from 0.95 to 0.99 (*Hervieux et al., 2014*). Similarly, the Parsnip experiment resulted in λ of caribou increasing from 0.95 to 1.02 (*Steenweg, 2011*; D Heard, 2016, unpublished data). These actions have not rapidly recovered caribou populations but the resulting trends are an improvement over the alternative, continued population declines. Formal caribou population comparisons from this study ended in 2013 because a maternity pen (a pilot trial of *in situ* captive breeding designed to increase calf survival) was initiated in the Columbia North subpopulation that year, which could confound results of the moose reduction. However, the 2017 caribou census for Columbia North revealed a count of 147 caribou, still higher than 2003, and 5 fewer than 2013, indicating at least stability from 2003 to 2017. The population effect of the maternity pen pilot trial was likely negligible (estimated to have added a net of 8 calves over 3 years; *Legebokow & Serrouya, 2017*), meaning that 14 years of population stability can likely be attributed to the moose reduction.

In our experiment, moose numbers were reduced using a change in hunting regulations, addressing a more ultimate cause of the apparent competition problem, rather than direct wolf control. Yet because moose were not reduced to the target developed by *Serrouya et al.* (*2011*; ∼300 moose), or even to the lower range of the predicted target (lower 95% CI: 167 moose), the caribou response was unlikely as strong as anticipated. Prior to the 1940s, moose were at even lower densities than predicted by *Serrouya et al. (2011)* and may have been absent from central and southern BC (*Spalding, 1990*; *Kay, 1997*; *Santomauro, Johnson & Fondahl, 2012*). The precautionary principle (*Doak et al., 2008*) would suggest reducing moose even lower than 300 to hold wolves at lower numbers. Recent wolf densities in winter are *c.* $13/1,000$ km$^2$, which is above a target developed by *Bergerud & Elliot (1986)*, who found that caribou mortality was offset by recruitment at the threshold of 6.5 wolves/1,000 km$^2$. Prey switching by wolves from moose to caribou could also be invoked to explain the lack of caribou population growth. However, an analysis of wolf scats suggests that prey switching did not occur (*Serrouya, 2013*), given that moose were gradually reduced over 10 years. When prey are reduced suddenly, prey switching becomes a greater risk (*Serrouya et al., 2015*) and can exacerbate caribou declines.

In most ungulate systems adult survival is necessarily high (*Gaillard, Festa-Bianchet & Yoccoz, 1998*), making statistical comparisons ($p < 0.05$) difficult (*McLellan et al., 1999*; *Hayes et al., 2003*). This pattern is also true of endangered species or low-density carnivores where large sample sizes are challenging to obtain. However, in the ecological literature there is increasing emphasis placed on stressing the magnitude of biological effects (*Burnham &*

*Anderson, 2002*). In our case, there was a substantial increase in caribou adult survival in the treatment area (∼8.5 units; Table 4), enough to considerably affect population growth (*DeCesare et al., 2012*; *Serrouya et al., 2017*). The same pattern occurred for other vital rates such as moose survival (7.5 units lower in the treatment area) and wolf dispersal (2.5 × greater in the treatment area), with *p*-values approaching statistical significance but the magnitude and direction of estimates supporting the primary hypotheses in this study. Caribou recruitment appeared unaffected by the moose reduction treatment, likely because bear predation, not wolves, is a major limitation on caribou calves (*Brockman et al., 2017*). Furthermore, calf recruitment in ungulates is highly variable (*Gaillard, Festa-Bianchet & Yoccoz, 1998*) and affected by many factors including spring weather (*Hegel et al., 2010*), so a reduction in wolves may be less likely to clearly affect this age class compared to adult survival, which is relatively unaffected by abiotic factors (*Gaillard, Festa-Bianchet & Yoccoz, 1998*).

Spatial and temporal variation in predation intensity has a major influence on population dynamics (*Creel & Winnie, 2005*). Predation on adult mountain caribou by wolves or cougars shifts from predominantly wolves in northern areas to cougars in southern areas (*Wittmer et al., 2005*). Bears were the second highest source of mortality in each of the southern and northern half of caribou range, but with both areas combined, they were the primary source of mortality. Furthermore, bears are a major predator of woodland caribou calves (*Adams, Singer & Dale, 1995*). *McLellan (2011)* and *Mowat, Heard & Schwarz (2013)* found grizzly bear densities inversely related to terrestrial meat in their diet and more closely linked to vegetative food, mostly fruit production (*McLellan, 2015*) and, grizzly bears appear to eat little meat in our study area (*Hobson, McLellan & Woods, 2000*). Because bear foraging is directed at vegetation, we did not expect a numerical response of bears to the moose reduction treatment. Cougar abundance and diets were monitored intermittently in the treatment area (*Bird et al., 2010*) using GPS cluster analyses (*Anderson & Lindzey, 2003*; *Knopff et al., 2009*), with moose comprising 5–43% of individual cougar diets (*Bird et al., 2010*). Cougar predation on caribou began following the peak and collapse of deer populations (*Serrouya et al., 2015*). After this dynamic, cougar numbers declined and wolf predation on caribou increased (*Stotyn, 2008*). These examples illustrate how conclusions drawn from landscape-level field experiments must consider how limiting factors change, often unpredictably, over space and time (*Doak et al., 2008*). Nonetheless, by combining what was observed in this and other case studies (*Courchamp, Woodroffe & Roemer, 2003*; *Wittmer et al., 2013*), some generalities are supported. In the context of apparent competition, high but especially fluctuating populations of primary prey can enhance extinction risk for rare prey (*Serrouya et al., 2015*). Maintaining lower and thus more stable populations of primary prey is expected to reduce predator switching and help maintain predators at low numbers (*Serrouya et al., 2015*). This generality is likely applicable to other caribou systems where moose are not the dominant ungulate biomass, but where other ungulates are expanding because of changing land uses and climates.

Wolves are highly mobile and fecund, so if their primary prey remain abundant during a period of wolf control, ingress occurs rapidly (*Ballard, Whitman & Gardner, 1987*; *Hayes et al., 2003*; *Mosnier et al., 2008*). Therefore, at least an 80% annual reduction in wolf

abundance is required to elicit a response in ungulate population growth (*Hayes et al., 2003*). However, if wolf control were to be implemented concurrently with primary prey reduction, it would likely have to be less intensive and less continuous relative to other areas where primary prey was not reduced (e.g., Yukon: *Hayes et al., 2003*; Quebec: *Mosnier et al., 2008*; Alaska: *Gasaway et al., 1992*; British Columbia: *Bergerud & Elliott, 1998*; Alberta: *Hervieux et al., 2014*).

In addition to this study, we are aware of three other attempts to reduce apparent competition by reducing primary prey: (1) the Parsnip study; (2) a study where 25,000 domestic sheep were reduced to 2,000 to try and recover endangered huemul deer (*Hippocamelus bisulcus)* in Patagonia (*Wittmer, Elbroch & Marshall, 2013*; *Wittmer et al., 2013*); and, (3) the removal of feral pigs to recover the island fox (*Urocyon littoralis*) on the Channel Islands of California (*Courchamp, Woodroffe & Roemer, 2003*), although periodic predator removal of golden eagles (*Aquila chrysaetos*) also occurred in that case. In a fourth study, a serendipitous experiment occurred when extensive poaching of African buffalo (*Syncerus caffer*) was linked to reduced lion (*Panthera leo*) numbers, resulting in a pronounced increase of impala (*Aepyceros melampus*; *Sinclair, 1995*). In the Parsnip study, results were similar to ours, with high wolf dispersal rates, and the caribou population stabilized. In the case of huemul, their decline was exacerbated as a result of increased predation by foxes (*Lycalopex culpaeus*) and pumas (*Puma concolor*), likely resulting from the abrupt decline in sheep and predators switching to huemul. The results of these manipulations suggest inconsistent responses to reducing primary prey as a recovery tool for species affected by apparent competition. This conclusion will complicate the decision-making process for management agencies, but evidence suggests that this approach should be considered when used in concert with tools that address other proximate and ultimate limiting factors.

Historical accounts (*Spalding, 2000*; *Seip, 1992*) suggest that mountain caribou were once at least an order of magnitude more abundant than they are today. During this period of caribou abundance, it is possible that the trophic interactions were reversed, and moose were the victim of apparent competition with wolf predation and harvest by First Nations (*Kay, 1997*) keeping them at low numbers. Extensive wolf control using poison and bounties occurred from 1906–1962 throughout BC (*McLellan, 2010*), and this along with climate and ecosystem change made it possible for moose to expand into southern BC. It is becoming increasingly evident that returning to the caribou-dominated system will require exceeding the biological population targets proposed by *Bergerud & Elliot (1986)* and *Serrouya et al. (2011)*. Random processes associated with small populations, and Allee effects due to predation (*McLellan et al., 2010*; *Armstrong & Wittmer, 2011*) will make it increasingly difficult to recover mountain caribou populations. However, an emerging pattern is that single management actions may halt declines, but multiple actions that address several limiting factors simultaneously will be required to achieve population growth (*Serrouya, 2013*; *Boutin & Merrill, 2016*).

## ACKNOWLEDGEMENTS

We thank the Selkirk College Geospatial Research Centre for providing the radio collars for moose survival estimation. Dave Mair of Silvertip Aviation piloted our telemetry flights and Clay Wilson of Bighorn Helicopters helped to catch most of our animals. Leo DeGroot, Cory Legebokow, Tara Szkorupa, Chris Ritchie, and John Surgenor helped with many logistical aspects. Kelsey Furk and Janice Hooge assisted with telemetry flights, and Dale Seip provided constructive comments on previous versions of this manuscript. We also benefitted from helpful discussions with Mark Hebblewhite, and an anonymous reviewer and Martin-Hugues St-Laurent provided insightful comments during the review process. This research is dedicated to the memory of our friend and teacher, Gary Pavan, who was instrumental to the completion of this project.

## APPENDIX 1. MOOSE POPULATION TREND COMPARISON BETWEEN THE TREATMENT AND REFERENCE AREAS

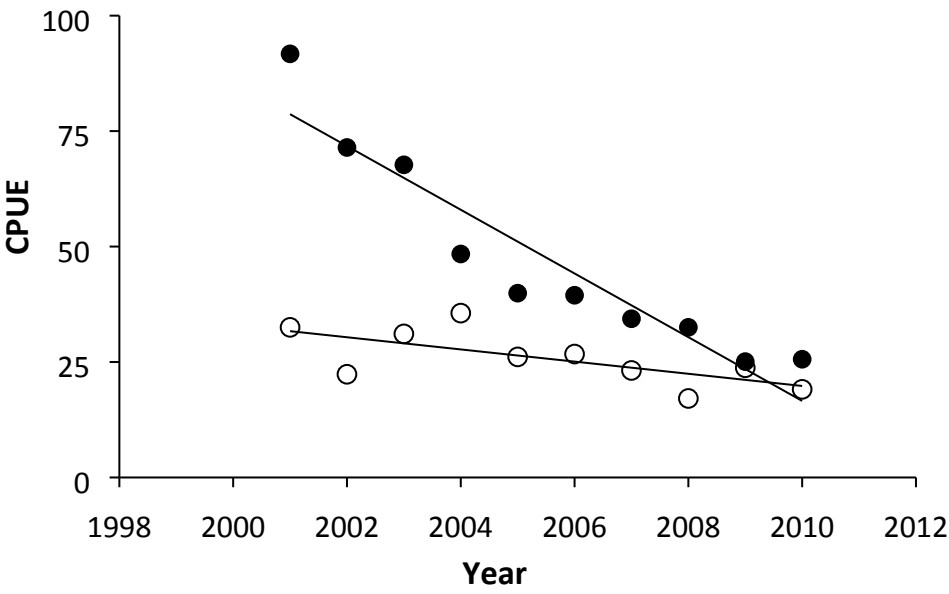

**Figure A1** **The slope of decline was more than five times greater in the treatment (solid circles) compared to the reference area (open circles; slopes were −6.88 [−9.02 to −4.68] compared to −1.32 [−2.46 to −0.26] for the respective treatment and reference areas; 1,000 bootstrapped iterations).** The slopes estimated here are slightly different from *Serrouya, McLellan & Boutin (2015)* because the reference area was larger to accommodate the wolf ranges and caribou subpopulations. Nonetheless, the ratio of the slopes between the two areas (i.e., comparing the magnitude of change between the treatment and reference) was similar when the smaller or larger reference area was compared to the treatment area.

### Funding

Funding to RS was provided by the Natural Sciences and Engineering Research Council of Canada, Alberta Ingenuity, and the Bill Shostak Wildlife Award. Funding for field components was provided by the Research Branch of the British Columbia Ministry of Forests, the Fish and Wildlife Compensation Program (Columbia Basin), Habitat Conservation Trust Foundation, the Simpcw First Nations, Downie Timber, Louisiana Pacific, Bell Pole, and the Revelstoke Community Forests Corporation. Radio collars for moose survival estimation were provided by the Selkirk College Geospatial Research Centre. The funders had no role in study design, data collection and analysis, decision to publish, or preparation of the manuscript.

### Grant Disclosures

The following grant information was disclosed by the authors:
Natural Sciences and Engineering Research Council of Canada, Alberta Ingenuity.
Bill Shostak Wildlife Award.
Research Branch of the British Columbia Ministry of Forests.
Fish and Wildlife Compensation Program (Columbia Basin).
Habitat Conservation Trust Foundation.
Simpcw First Nations, Downie Timber, Louisiana Pacific, Bell Pole, and the Revelstoke Community Forests Corporation.

### Competing Interests

The authors declare there are no competing interests.

### Author Contributions

- Robert Serrouya conceived and designed the experiments, performed the experiments, analyzed the data, contributed reagents/materials/analysis tools, wrote the paper, prepared figures and/or tables, reviewed drafts of the paper.
- Bruce N. McLellan conceived and designed the experiments, performed the experiments, contributed reagents/materials/analysis tools, wrote the paper, reviewed drafts of the paper.
- Harry van Oort performed the experiments, analyzed the data, contributed reagents/materials/analysis tools, wrote the paper, prepared figures and/or tables, reviewed drafts of the paper.
- Garth Mowat performed the experiments, contributed reagents/materials/analysis tools, wrote the paper, reviewed drafts of the paper.
- Stan Boutin contributed reagents/materials/analysis tools, wrote the paper, reviewed drafts of the paper.

### Animal Ethics

The following information was supplied relating to ethical approvals (i.e., approving body and any reference numbers):

Captures adhered to BC Provincial Government and University of Alberta animal care protocols (permit # VI08-49757, and 690905, 2004-09D, 2005-19D).

## Data Availability

The raw data and code have been provided as Supplemental Files.

## Supplemental Information

Supplemental information for this article can be found online at http://dx.doi.org/10.7717/peerj.3736#supplemental-information.

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
