# Peer review of "Experimental moose reduction lowers wolf density and stops decline of endangered caribou"

_PeerJ, doi:10.7717/peerj.3736_

## Round 0.1 · original submission · Minor Revisions

Dear Dr. Serrouya,

Your manuscript has been reviewed by two experts in the field, and overall the reviews have been positive. However, both reviewers highlight revisions required to improve the manuscript before being acceptable for publication with PeerJ.

Both reviewers highlight that some of the inferences made are vulnerable to small sample sizes. Despite this being understandable given the nature of the study, the risks in terms of power and inference should be highlighted more explicitly in the discussion.

In the abstract, there is an undefined abbreviation - please write out the full title of the design (before and after control impact (BACI)).

Overall, I suggest that you address, or rebut, the extensive comments from both reviewers (see below).

Kind regards

Andrew

·

Basic reporting

This manuscript is clear, well written, efficiently structured, supported by several peer-reviewed references, with clear and relevant hypotheses.

More details provided in the section "General comments for the authors" below.

Experimental design

The experimental design is astonishing, probably one of the rare lansdcape-level field experiment conducted to support caribou recovery in the world. The research question is well defined, relevant and meaningful. Ethical standards could be more supported by providing Animal Welfare Certificate # in the manuscript for the collared animals. Methods are sufficiently described to be understandable.

More details provided in the section "General comments for the authors" below.

Validity of the findings

Very interesting results, robust data even if the sample size is low (but how difficult to obtain), conclusions are okay and supported by the results (except lines 609-619, more speculative).

More details provided in the section "General comments for the authors" below.

Additional comments

I really enjoyed reading this interesting manuscript about one of the most impressive landscape-level field experiment conducted to support woodland caribou recovery. Now that most caribou biologists are confronted with a global decline of Rangifer all around the northern hemisphere, thinking outside the box and testing new conservation strategies and management actions is greatly needed. Based on this, I am convinced that the topic and questions addressed in this study are highly relevant. I’m dealing with that kind of questions in eastern Canada and sincerely believe that this manuscript deserves publication. The manuscript is interesting, well-written and supported by rigorous data and collected over a huge study area for several years, which is very impressive, and the statistical design is interesting and robust, despite some limits imposed by the sample size.
I’ve listed below some minor points/comments that could help clarify some parts of the manuscript before publication.

- Caribou recruitment: at the end of the abstract (line 36), it is written that “Caribou recruitment was unaffected by the treatment”. Interesting, but one more sentence is needed to explain potentially why. In the discussion, it’s also not detailed (i.e. only one short sentence; see lines 557-559), although it could be an important driver explaining caribou decline or stability, or even short-term increase, being highly variable between years in ungulates (Gaillard et al. 1998).

- Line 158: Report the ratio (~15:1) for the pellet group counts, it’s easier to understand.

- Line 187: Animals monitored every 2-4 weeks from fixed-wing aircraft are VHF collared? If so, precise.

- Table 1: Explain in the caption what does Y and N mean; it’s yes or no, but it could be written (yes, no) instead of Y and N. Also, what could be the impact of not following abundance, trend and recruitment of wolves in the reference areas on the results and conclusions? It should appear in the discussion, briefly.

- Lines 303-313: Wolf recruitment was assessed with motion triggered cameras, which is interesting, but you could have done more, i.e. estimate abundance or density, following random encounter models (Rowcliffe, J.M., J. Field, S.T. Turvey & C. Carbone. 2008. Estimating animal density using camera traps without the need for individual recognition. Journal of Applied Ecology. 45:1228-1236) or spatial presence-absence models (Ramsey, D. S. L., P. A. Caley, and A. Robley. 2015. Estimating population density from presence-absence data using a spatially explicit model. Journal of Wildlife Management 79:491-499). I’ve used these two models recently, and if you have enough cameras, it could be easily done. Note for the editorial board: This comment does not prevent publication in PeerJ (!!), it’s just an interesting avenue to consider in the future.

- Lines 322-327: Is there a bias comparing these two methods to assess lambda?

- Lines 388-390: When human-caused mortality rates are confounded (roadkill, trapping and hunting), is there a statistical difference between the treatment and the reference areas?

- Line 424: It is not the subpopulation that increased following reduction, it’s the growth rate. I had hard time finding the “increase” in population abundance (y-axis in Fig.4) but understood that you were talking about an increase in lambda. Please adjust.

- Figure 5: Could it be possible to draw regression lines or before/after boxplots to compare the % of calves before and after? It seems that a regression line will suggest opposite trends (with a lot of variation) that might not be statistically significant.

- Discussion: It could have been appreciated to use subtitles in the discussion, to orient the reading and organize the ideas. For examples, lines 547-559 are very interesting, and draw the limits imposed by a small dataset, but are important and could be grouped (along with some other sentences) in a section.

- Line 504: What about other systems, when predators are less “specialized” on the more productive ungulate species or when prey are more diversified? Working in the Atlantic-Gaspésie caribou population, with coyotes and black bears but no wolves, I am more than interested in reading about how such results could be inferred to different systems.

- Lines 542-546: Could it be possible that wolves are switching from moose to caribou in the treatment populations, limiting the potential for an increase in abundance? Do you have any fecal pellet samples collected before and after the treatment to figure out if the caribou-to-moose ratio in the feces has changed?

- Lines 582-583: Passing quickly from high to low moose abundances could trigger prey switching, no? If moose populations are high, and we decrease them quickly, than the primary prey population is not stable, which is not expected to reduce predator switching based on this sentence… it’s confusing and could be detailed more.

- Lines 609-619: This section is interesting, but more speculative. I understand that the authors weren’t interested in creating high expectations so this information is needed, but this is maybe too long.

- Appendix 1: In the caption, a reference is made to Chapter 5, but I cannot find Chapter 5 in this manuscript. Rob, this caption is extracted from your PhD dissertation, page 212…! Please provide the appropriate reference ; )

Reviewer 2 ·

Basic reporting

This paper reports on an interesting and relevant research area aiming to answer the question whether reduction of a primary prey species can reduce predation on a secondary prey species, a process known as apparent competition. The authors take advantage of a large scale manipulative experiment in where the primary prey of wolves is limited by hunting harvest in an experimental area and the results are compared with data from a control area without reduction.

The ms is professionally written with adequate references and with a good structure.

There are a number of points that should be addressed to make the paper clearer and to allow the reader to better evaluate the results achieved and the conclusions made. I will list those points below.

Experimental design

The study design is relevant to the question asked and should be within the aims and scope of the journal. Research questions are well defined and and the topic is relevant and deservs further attention in ecology and management. Investigations are performed with adquate methods and with high technical and etical standards.

Validity of the findings

The authors view their results as giving support for their main hypothesis that a reduction in population size of the predators primary prey (moose) results in a lowered density of the predator population and that this leads to a lower predation on the secondary prey species (caribou).
The results are to some extent contradictory which make it difficult to draw clear conclusions about if the main hypothesis could be confirmed or rejected. I believe this is to some extent stemming from low sample sizes for a number of parameters investigated.
Summarizing the results that supported their main hypothesis was that there was a clear effect of moose hunting on population development although moose survival was not significantly different between areas. Also caribou survival increase in the treatment area after moose decline but declined in the reference area. Dispersal rates of wolves were higher in the treatment area but this result need to consider age of dispersers because dispersal is strongly related to age in wolves.
Results not supporting the main hypothesis was that, two of three caribou populations continued to decline after moose reductions in the experimental area, mortality rates of wolves did not differ among areas and there was no change in caribou recruitment among areas. In addition, recruitment of wolves between high and low moose density area not comparable due to small sample size and there was no data on wolf abundance, density and trend between areas.
Overall I think the study is contributing to our overall understanding of predator-prey ecology and apparent competition and the authors have done a good job compiling a diverse dataset to test an interesting and relevant research question for management.

Additional comments

Minor points.
L 35. I believe this is annual estimates of survival and should be specified.
L 66-67. Give reference for this statement.
L 103. This statement suggests that other factors may be more or at least equally important for liming caribou - what are these and is there any quantitative data that support this conclusion?
L 167. The Frisby-Quest area seems to be situated both in the experiment and in the control area. How did you deal with this?
L 200-201. Why was not wolf abundance trend and recruitment estimated in both areas? Is there any evidence for assuming similar population density among areas?
L 204-205. Is it possible that lower harvest just reflected a lower effort therefore were not reflective of density? If not, why?
Table 1. There seem to be some important demographic information missing for the wolf population in the reference area which makes it partly difficult to evaluate one important response parameter to the reduction in moose numbers.
L 305-306-. The estimate of wolf recruitment has a very small sample size and will be strongly exposed to random variation among packs and years which means that firm conclusions from this response variable cannot be made. I’m also hesitant to the method for estimating the minimum number of pups for each pack and how reliable this method may be to actually capture the true number of pups.
L 370-371. Dispersal is usually strongly age dependent in mammals so presenting just dispersal rates without accounting for potential age differences among dispersers may be misleading.
L 464-481. That the moose population was reduced as a response to increased harvest was supported but a trivial result in itself.
Line 482-. There is not strong support for that the wolf population declined as a result from moose population decline. Authors bring up several mechanisms but these differences are not supported by statistical analyses although I realize that the small sample sizes used will reduced the power of these analyses.
Line 511-. The authors claim that their final hypothesis, that reduced wolf numbers would lessen the rate of caribou decline relative to spatial and temporal contrasts, was partially supported. However, viewed in terms of how the different subpopulations responded between areas I think caution is needed when making conclusions.
L 520-. The authors argue that…..”It is tempting to conclude that the experiment was not a success because caribou population growth was not immediate and pronounced in the treatment area. However, our results are similar to other management actions intended to recover declining caribou herds.” I think that the reasoning that the results from the current study were similar to the results from other management actions elsewhere does not justify that the results from the current study are viewed in a different light.
L 586-. The authors claim (by reference) that “….at least an 80 % annual reduction in wolf abundance is required to elicit a response in ungulate population growth.” This sounds like a very high reduction is needed to result in positive population growth of prey and if true this would seem contradictory to the expected outcome of this study. Do the authors mean that wolf control is needed in addition to reduction in the main prey to have an effect on caribou population growth? I think the authors need to explain how these numbers relate to the results in the current study.

L 879. The authors refer to Chapter 5 in the text but does not give a reference for this.

---

## Round 0.2 · Minor Revisions

Dear Dr. Serrouya,

I invited one of the original reviewers to assess your responses to their comments, however the reviewer did not respond.

I have read through your responses to reviewers and my original comments. While you have addressed many concerns very well, there remains three minor revisions which I believe you should consider:

Line 529: Replace 'demonstrated' with 'demonstrate' (..demonstrate an improved demographic…)

Line 549: Due to uncertainty, I suggest you insert 'potentially' - 'stability can potentially be attributed to the moose reduction'

Title: On reflecting on your results and experimental design, I would be of the opinion that the title asserts too strong an inference. There is the possibility causation has not been established with high degree of certainty.

I suggest that you consider a retitle: e.g. “Experimental moose reduction IS ASSOCIATED with lower wolf density and the halting of the decline of endangered caribou”

Kind regards

Dr. Andrew Byrne

---

## Round 0.3 · accepted · Accept

Dear Dr. Serrouya,

I have reviewed your rebuttal letter and will accept your arguments with regards to the justification of the title in this instance. I concur that it is extremely challenging and expensive to undertake such experiments. Though I would point out, for your awareness, large-scale replicated wildlife removal experiments have been carried out for disease control investigation (e.g. badger-TB episystem in the UK and Ireland ; Griffin et al. 2005 PVM; Donnelly et al. 2006 Nature), which does reduce the risk of incorrect inference, and improves the robustness and transferability of results to other geographic locations. Despite this, I would like to congratulate you and your team on an impressive piece of work.

Kind regards

Andrew